# Strategies for Functional Interrogation of Big Cancer Data Using Drosophila Cancer Models

**DOI:** 10.3390/ijms21113754

**Published:** 2020-05-26

**Authors:** Erdem Bangi

**Affiliations:** Department of Biological Science, Florida State University, Tallahassee, FL 32306, USA; ebangi@bio.fsu.edu

**Keywords:** cancer, Drosophila, big data, cancer genomics

## Abstract

Rapid development of high throughput genome analysis technologies accompanied by significant reduction in costs has led to the accumulation of an incredible amount of data during the last decade. The emergence of big data has had a particularly significant impact in biomedical research by providing unprecedented, systems-level access to many disease states including cancer, and has created promising opportunities as well as new challenges. Arguably, the most significant challenge cancer research currently faces is finding effective ways to use big data to improve our understanding of molecular mechanisms underlying tumorigenesis and developing effective new therapies. Functional exploration of these datasets and testing predictions from computational approaches using experimental models to interrogate their biological relevance is a key step towards achieving this goal. Given the daunting scale and complexity of the big data available, experimental systems like Drosophila that allow large-scale functional studies and complex genetic manipulations in a rapid, cost-effective manner will be of particular importance for this purpose. Findings from these large-scale exploratory functional studies can then be used to formulate more specific hypotheses to be explored in mammalian models. Here, I will discuss several strategies for functional exploration of big cancer data using Drosophila cancer models.

## 1. Introduction

Cancer is a genetically complex and diverse disease. Most tumors carry alterations in a large number of cancer driver genes and individual tumors of the same type exhibit great diversity in their genomic landscapes [1]. Tumor initiation and progression is also accompanied by dynamic, systems-level changes at the level of transcription, translation, posttranslational modifications, chromatin, and DNA methylation patterns. In recent years, genome-wide omics datasets documenting these global changes from large numbers of tumors of different types have become available.

Sophisticated computational and statistical analyses of these datasets routinely identify specific molecular signatures that correlate with different tumor phenotypes, revealing common and unique features of tumors with different genomic landscapes or tissues of origin. While these correlative studies have been instrumental in identifying candidate biomarkers to predict disease outcome and drug response, establishing causal relationships between specific molecular alterations and tumor phenotypes or response to therapy is difficult to achieve computationally. Functional studies are essential to test predictions from these studies in experimental models in a comprehensive, systematic way to determine their relevance in tumor initiation and progression and to identify novel potential drug targets and other actionable vulnerabilities.

Over the past hundred years, Drosophila—with its powerful genetic tools and practical advantages—has opened the door to understanding fundamental aspects of development, cell biology, and signal transduction. In addition to its contributions to basic science, Drosophila also has a strong track record as a useful disease model and a drug discovery platform [2]. Around 75% of disease associated human genes have Drosophila orthologs. In addition, most disease relevant molecular pathways are highly conserved and thoroughly characterized in flies. Many complex disease states including cancer have been successfully modeled in flies in a whole animal setting. An arsenal of sophisticated genetic tools that represent over a century of use and refinement provides the ability to generate complex genetic combinations that reflect the complexities of human disease [3].

The history of Drosophila cancer research is rich and almost as long as the history of Drosophila as a model system itself [3]. It starts about a hundred years ago with the first reports of tumors in flies and continues on with many key discoveries along the way, including the first tumor transplantation assays in the 1960s, studies of mechanisms of growth control, cell–cell interactions, and cell competition in developmental contexts that provided key insights into tumor biology, culminating in fly models of cancer genetically engineered to carry specific cancer associated mutations.

These latter models were generated either by genetically manipulating Drosophila orthologs of genes altered in tumors (e.g., cancer driver genes) or directly introducing wildtype or altered versions of human genes into flies. Some models of cancer have targeted the Drosophila equivalent of the affected tissue in humans (e.g., intestine or the brain), while others used generic epithelia such as the imaginal discs where cancer relevant signaling pathways are highly conserved. Both approaches have been successful in capturing key hallmarks of tumorigenesis including overproliferation, evasion of apoptosis and senescence, epithelial–mesenchymal transition, migration, invasion, aspects of metastasis, as well as drug response.

An extensive review of the Drosophila cancer literature is not the intended goal of this article; several excellent reviews that summarize contributions of Drosophila to cancer research are already available [3,4,5,6,7]. Instead, I will discuss how we can build upon our previous successes in cancer research by effectively leveraging sophisticated genetic tools and the practical advantages of our model system for functional exploration of big cancer data. These strategies would allow us not only to use Drosophila cancer models to extract biologically relevant information from these large datasets, but also to use big cancer data to improve our models.

## 2. Functional Validation of Genomics Datasets

As massively parallel sequencing technologies became cost effective and, as a result, more widespread, the first tumor genome profiling efforts to emerge were those that sought to catalogue genetic alterations in tumor cells by comparing normal and tumor DNA sequences. These tumor sequence datasets still remain to be the most abundant big data in cancer research [8]. These genomic approaches typically utilize DNA sequencing to identify mutations—both somatic and germline—and chromosomal rearrangements in tumors. They routinely identify deleterious, functionally validated variants in well-established cancer driver genes (oncogenes and tumor suppressors). However, such variants represent only a small fraction of tumor genome landscapes; the majority of variants observed in sequenced tumors are either novel variants with unknown functional impact or in genes with no known roles in tumorigenesis.

Most of the novel and unique variants observed in human tumors will represent passenger mutations that perhaps reflect the unique mutational background of the cell from which the tumor originated or mutations that accumulated over the course of tumorigenesis that do not have any functional relevance. However, some may represent yet undiscovered, low frequency cancer driver genes or may impact tumor progression and drug response in subtle, context-dependent ways. In addition, DNA sequencing and copy number variation (CNV) analyses have revealed recurrent copy number changes associated with many tumor types, which usually include loss or amplification of chromosomal regions that may include dozens or hundreds of genes in some cases. For instance, sequencing studies have identified approximately 30 recurrently mutated, functionally validated cancer driver genes in colorectal tumors at the population level [9,10]. Of the 60–120 somatic mutations typically found in an individual colon tumor, only 3–6 are in known drivers; the rest are novel variants with unknown functional significance [11,12].

In summary, a common theme in big cancer data is the presence of large numbers of candidate genes or variants whose biological relevance is unknown. While sometimes, it is possible to prioritize candidate genes based on their functions, unbiased approaches for experimentally evaluating the impact of such genes using disease phenotypes would be useful to identify novel drivers of tumorigenesis and candidate drug targets. A genetic modifier screen approach could be modified to test a large number of such genes by genetically manipulating their Drosophila orthologs in tumor models with different genetic contexts. This is a scalable approach that can be adapted for both gain and loss of function studies and utilizes publicly available large-scale transgenic and mutant collections. Even though recurrently altered cancer drivers make up a small fraction of the tumor genome landscape, they are excellent starting points to build cancer models that can then be used to functionally interrogate the roles of the remaining uncharacterized genomic alterations observed in tumors to determine their potential contributions to tumorigenesis (Figure 1).

Given the large number of known cancer driver alterations observed in tumors [1], even the first-generation cancer models that focus solely on well-established cancer drivers need to be genetically complex. Sophisticated genetic tools available in Drosophila provide a unique opportunity to build and study large numbers of such models that reflect these complex tumor genome landscapes. We started exploring this strategy several years ago by building a panel of colorectal cancer models based on tumor sequence data generated by The Cancer Genome Atlas (TCGA) [9,10]. These first-generation models only included recurrently mutated genes with well-established roles as drivers of tumorigenesis. The most complex model in this collection targeted five drivers of colorectal cancer. We then developed a multigenic vector platform that allows consolidation of multiple transgenes into a single construct, which can be inserted into a single genomic location [11], minimizing the need to bring together multiple constructs using standard genetic crosses. Using this platform, we were able to build more complex models that targeted 8–13 genes, which allowed us to also incorporate alterations in genes that are not recurrently mutated in human tumors and those with no known roles in tumorigenesis into our models. By simplifying the genetic background of the cancer model, this approach also makes it easier to perform additional genetic interaction experiments or genetic modifier screens using these models.

Genetic screens that use complex, in vivo phenotypes as read-outs are a key strength of Drosophila as a model system. As previously discussed, many aspects of human cancer including proliferation, apoptosis, senescence, epithelial–mesenchymal transition, migration and dissemination have been captured in Drosophila models. Genetic and compound screens that use disease relevant phenotypes, imaging, or luciferase-based methods have been reported [13,14,15]. Lethality is another commonly used read-out for large-scale experiments and screens [16,17] that can be calibrated to detect both enhancers and suppressors. All of these phenotypes can be adapted for genetic screens designed to functionally interrogate tumor genome data.

For instance, a Drosophila model of glioblastoma established by co-activating Epidermal Growth Factor Receptor (EGFR) and Phosphatidylinositol-3-Kinase (PI3K) pathways—commonly observed cancer driver alterations in these tumors—in glial cells and their precursors [18,19] can be used to test potential roles of other mutated genes observed in sequenced glioblastoma tumors in a genetic modifier screen. A lethal phenotype established by targeting the cancer driver alterations to the developing larval brain could be used as a primary screening read-out. Modifiers of the lethal phenotype can then be explored in more detail using secondary assays to investigate how they impact different hallmarks of tumorigenesis. Such tumor phenotype-based assays have been reported for both larval and adult models of glioblastoma [19,20]. These studies could identify novel cancer driver genes, candidate drug targets, and druggable nodes that can be followed up in mechanistic studies in both Drosophila and mammalian models.

Exploring roles for novel genes or variants observed in tumors requires introducing additional genetic manipulations into these already complex genetic backgrounds. Furthermore, such studies need to be large in scale to be able to explore the large amount of sequencing data available. Publicly available tools, resources, and practical advantages like low cost and speed allow functional studies in Drosophila at a scale that is not feasible in most genetically tractable model systems. For instance, several large-scale transgenic Drosophila resources that allow tissue specific and temporally controlled manipulation of almost every gene in the Drosophila genome are available [12,13,14,15]. These collections could be used for a systematic and comprehensive functional exploration of alterations observed in human tumors by genetically manipulating their Drosophila orthologs. Several small scale studies that have utilized Drosophila and other genetic model systems for functional validation of GWAS variants [16,17,18,19] and exploration of the functional impact of novel disease associated genes or variants [15,20,21,22] can serve as proof-of-concept studies for such large-scale comprehensive efforts.

A related challenge is evaluating the functional impact of novel variants on protein activity. The vast majority of variants identified in tumor sequence studies are novel and do not fall into previously identified hotspot locations. Functional studies are required to determine whether these variants are deleterious and how they impact protein function (e.g., gain of function, loss of function, neomorphic). While there are excellent functional prediction algorithms that can be used to determine the likelihood that a mutation is deleterious [21,22,23] and prioritize novel variants, these predictions ultimately need to be experimentally validated.

There are a number of widely available gene editing and transgenic methods in Drosophila that can be scaled up to explore the functional impact of novel variants identified by sequencing studies. For instance, in cases where the particular mutated residue is conserved, the corresponding mutation in the Drosophila ortholog can be generated by the gene editing tool CRISPR/Cas9. Tissue-specific or conditional knock-out studies can be used to gain insights into the biological functions of such genes. As most variants observed in human tumors are heterozygous, large-scale genetic interaction experiments can be performed by introducing candidate variants as heterozygous alleles into cancer models to test their ability to modify tumor phenotypes. In addition to determining the functional relevance of novel variants in cancer, these studies can also provide experimentally validated datasets to improve the performance of functional prediction algorithms.

In cases where the specific mutated residue is not conserved in the Drosophila ortholog, it is possible to directly introduce the mutant version of human genes observed in tumors into flies, while simultaneously knocking down or knocking out the Drosophila ortholog. Transgenic flies carrying the wildtype versions of these genes can be used as controls. This “humanization” strategy has been previously reported as a tool for functional exploration and characterization of variants observed in rare human genetic disorders [24,25] and may also work for some genes that do not have clear Drosophila orthologs. The multigenic vector technology we recently reported [16] can also be adapted to co-express the human variant and a series of short hairpins targeting the Drosophila ortholog in a single construct. While some of these latter strategies require more sophisticated technologies and may be more challenging, especially if the cancer model being used is already genetically complex, it is still possible to scale up this approach for functional exploration of large numbers of novel variants observed in sequenced human tumors.

A similar approach can also be used to model gene-fusion events that produce truncated proteins or novel fusion products with inappropriate or neomorphic activities. Such fusion events are being increasingly identified as recurrent drivers of tumorigenesis, especially by RNA sequencing studies [26]. Drosophila cancer models generated by directly expressing human fusion genes have already been reported [27,28]. These studies can be further expanded to other recurrently observed human fusion genes to create cancer models that better reflect human tumor genome landscapes. They can also be used to systematically test novel fusion events with unknown functional relevance. The relatively low cost and rapid nature of transgenesis in Drosophila makes it feasible to build additional transgenic lines that express the full-length version of each human gene as well as each half of the fusion product as truncated proteins as controls to investigate the mechanism of action of the fusion product.

Finally, Drosophila also offers a platform for the exploration of any potential non-autonomous roles for germline variants in tumorigenesis. Each human tumor develops in a different germline background, with a large number of rare, potentially deleterious, and mostly heterozygous germline mutations; how these germline mutations interact with the somatic tumor genome landscape during tumorigenesis is not well-studied. A small number of these variants represent deleterious mutations in susceptibility genes that require a somatic mutation to inactivate the wildtype allele to initiate tumorigenesis, as proposed by Knudson in his two-hit theory of cancer causation [29,30]. Such variants are relatively rare and can easily be distinguished from germline variants that remain heterozygous in tumors by comparing sequence data from normal and tumor samples. Even though heterozygous germline variants are not likely to be key drivers of tumorigenesis alone, the overall unique germline context in which a tumor forms can influence its behavior and drug response.

A functional exploration of how germline and somatic mutations interact during tumorigenesis is also complicated by the fact that, while somatic alterations are only present in tumor cells, germline alterations are found in both the tumor cells and their wildtype neighbors. Therefore, in addition to cell autonomous interactions with somatic mutations in tumor cells, germline mutations can also affect tumorigenesis in a cell non-autonomous fashion by altering the interactions between tumor cells and their non-tumor neighbors. These interactions can be captured and explored by introducing mutant alleles for Drosophila orthologs of candidate germline variants observed in human tumors while inducing somatic alterations in a tissue specific manner to establish tumors in these germline backgrounds. It is also possible to use orthogonal or intersecting expression systems [31] to induce genetic manipulations corresponding to germline alterations ubiquitously and those corresponding to somatic alterations in a tissue-specific manner.

In addition to genomic alterations and chromosomal abnormalities, global changes at the transcriptional, translational, post-translational, and epigenetic levels have been documented in tumors [23]. Sophisticated computational approaches have been developed to mine these large datasets and identify specific signatures that correlate with different patient outcomes and drug response [24,25]. While such correlative studies are excellent starting points to extract biologically meaningful information from big cancer data, they cannot provide direct insights into functional contributions of these changes to tumor initiation, progression, and drug response. Building more representative cancer models that capture as much of the complexity and diversity of human tumor genome landscapes as possible can significantly increase the utility and predictive power of Drosophila as a platform for functional exploration of other big cancer data. In the next sections, I will discuss how such models can be used in genetic modifier screens to test the biological relevance of changes observed in tumors in a systematic, comprehensive way (Figure 2).

## 3. Functional Exploration of Tumor Transcriptomes

In recent years, gene expression datasets from thousands of patient tumors have also become available and led to a large number of gene expression signatures that correlate with patient outcomes and treatment responses. Such transcriptomic signatures have enormous potential as biomarkers in personalized cancer medicine and functional exploration of their roles in cancer could have a significant impact on our understanding of molecular mechanisms underlying tumorigenesis, in a way, providing “mechanisms of action” for these complex, multigenic molecular signatures. Given their importance as predictors of patient outcome, one or more of the genes in these signatures can also be promising drug target candidates. A systematic and comprehensive functional validation of candidate biomarkers can be challenging in vertebrate genetic models.

Large transgenic Drosophila collections for gene overexpression and knock-down [32,33,34,35] provide an opportunity to explore the biological relevance of large numbers of gene expression changes reported in human tumors. Drosophila orthologs of genes upregulated and downregulated in tumors can be overexpressed and knocked-down, respectively, in Drosophila cancer models in genetic modifier screens. Both organismal lethality and specific tumor phenotypes have been used in large-scale genetic and drug screens in Drosophila [36]. The availability of a large number of assays that measure key aspects of tumorigenesis such as proliferation, apoptosis, senescence, epithelial–mesenchymal transition, migration, metastasis-like phenotypes, and metabolic changes allows a comprehensive analysis of roles for such genes in tumorigenesis in follow-up studies and can provide more specific hypotheses to test in mammalian cancer models where such large-scale, exploratory studies may not always be as practical.

In addition to changes in the expression levels of protein coding genes, sequencing studies also revealed global changes in other types of RNA in tumor cells, microRNAs, and long noncoding RNAs (lncRNA) in particular [8,37]. These cancer associated RNAs—sometimes referred to as oncomiRs and onco-lncRNAs—can have global effects on gene expression and signaling pathway output, affecting a wide number of cancer relevant cellular processes, often in a tissue specific manner [8,38,39,40,41,42]. Drosophila cancer models can be used to explore functional relevance of genes and pathways dysregulated by these RNAs on tumorigenesis in a comprehensive and systematic way to provide more specific hypotheses regarding how they contribute to tumor progression.

For instance, miR-21 is one of the most frequently upregulated microRNAs in tumors and serves as a useful prognostic biomarker for many tumor types [43]. miR-21 has a large number of target genes [44]; its upregulation in tumors leads to the downregulation of many genes. As a result of its broad and tissue-specific effects on gene expression, it has been difficult to establish a clear mechanism of action for miR-21 as an oncomiR. miR-21 target genes can be knocked down systematically in Drosophila cancer models to evaluate their ability to modify tumor phenotypes and identify which miR-21 targets are most relevant for its oncogenic activity. Drosophila models can also help explore context and tissue dependent differences in its mechanism of action by exploring the roles of miR-21 targets in different cancer models (i.e., comparing colorectal cancer models with different genetic makeups or comparing a glioblastoma model to a colorectal cancer model).

## 4. Functional Exploration of Proteomic Datasets

Proteins are the final translational products of mRNA transcripts that mediate a variety of key cellular processes. As a result, alterations in protein levels and activity are key mediators of tumor initiation, progression, and drug response. Proteins are also the targets for the vast majority of targeted therapies in the clinic or under clinical development. As protein stability and activity is regulated by a wide number of posttranslational mechanisms, the correlation between protein expression levels and corresponding mRNA levels is relatively low [45,46,47]. In recent years, several large-scale efforts have emerged to profile tumor proteomes and catalogue differences in global post-translational modifications like protein phosphorylation between tumor and normal cells [8,48,49,50]. These datasets complement genomic datasets by identifying proteins with altered activity without any corresponding genomic alteration and hold great promise as candidate drug targets and biomarkers.

As is the case for most big cancer data, these studies typically identify a large number of changes in tumors. Identifying alterations biologically most relevant to cancer through functional studies could help streamline drug discovery efforts by prioritizing candidate drug targets for drug development. The approaches I discuss in the previous sections can also be applied for large-scale functional exploration of these datasets by genetically manipulating Drosophila orthologs of genes encoding proteins with altered expression or activity levels in tumors. These approaches could include classic modifier screens performed by removing a functional copy of each gene by bringing in loss of function alleles, transgenic constructs that ectopically express wildtype, dominant negative, constitutively active versions of these proteins and long or short hairpin constructs for RNA interference (RNAi) mediated knockdown in Drosophila cancer models. Genes that alter tumor phenotypes can then be further explored in Drosophila to obtain mechanistic insights and formulate hypotheses that can be tested in mammalian models.

## 5. Functional Exploration of Epigenomic Datasets

Epigenomic alterations reported in tumors include global or gene-specific changes in DNA methylation and histone modification patterns [8]. Some of these epigenetic changes directly impact genes with known cancer relevant functions, leading to loss of tumor suppressor gene expression or inappropriate expression of oncogenes. Some have also been used as biomarkers of drug response; for instance, epigenetic silencing of a DNA repair enzyme by DNA methylation of its promoter has been reported as a predictor of response to treatment with a DNA alkylating agent [51]. However, such well characterized epigenetic drivers of tumorigenesis represent a small fraction of reported tumor epigenome landscapes. The vast majority of changes documented by these studies—even those that correlate with specific patient outcomes—remain functionally unexplored.

Posttranslational modification of histone proteins is a key epigenetic mechanism that governs regulation of gene expression and is maintained by an array of chromatin remodeling complexes [52]. Alterations in chromatin states observed in tumors can sometimes be linked to specific alterations in genes encoding chromatin remodeling proteins themselves. However, because of the genome-wide and context-dependent effects of chromatin remodeling proteins on gene expression, clearly defining specific tumor promoting pathways downstream of mutations in chromatin remodeling genes is difficult [53]. For instance, both gain and loss of function mutations in chromatin remodeling genes have been reported in tumors, suggesting that they can be oncogenic or tumor-suppressive, depending on the specific cellular and tissue context. It is also important to note that, in some cases, epigenetic changes in tumors are secondary consequences of mutations in other cancer driver genes that alter signaling pathway outputs or dysregulate key biological processes, rather than being direct results of specific genomic alterations in chromatin remodeling genes. As a result of the dynamic, multifactorial nature of the chromatin remodeling process itself combined with the broad and pleiotropic effects of epigenetic alterations on biological processes, it can be challenging to predict which changes observed in tumor epigenomes are functionally relevant.

The general relationship between chromatin state and gene expression as well as genes encoding key chromatin remodeling complexes are highly conserved in Drosophila [54]. Potential roles for alterations observed in genes encoding chromatin remodeling proteins can be tested by genetically manipulating their Drosophila orthologs in cancer models, as described in the previous section. Given the broad and context-dependent effects of chromatin remodeling on gene expression, cancer models that target the corresponding or functionally equivalent tissue (brain, gut, and so on) may be particularly helpful in these studies. It may also be useful to explore global effects of such variants on epigenomes of Drosophila models to explore if patterns of global epigenetic changes can be recapitulated. Assays that have been developed to assess heterochromatin states in Drosophila [55] can also be helpful to evaluate global epigenetic changes induced in cancer models as a result of these genetic manipulations.

A complementary and perhaps more useful approach would be to focus on exploring the functional consequences of epigenomic changes observed in human tumors rather than trying to replicate these patterns in Drosophila cancer models. This could be achieved by targeted genetic modifier screens informed by big epigenome data, where Drosophila orthologs of genes that are epigenetically silenced or activated in human tumors are genetically manipulated in Drosophila cancer models to evaluate their effects on tumor phenotypes. Hits from these screens can then be tested in various combinations observed in human tumors to explore functional effects of more complex multi-gene epigenomic signatures.

DNA methylation—especially of cytosines in CpG dinucleotides—is an essential mechanism for controlling gene expression and regulates many cancer-relevant processes [56,57]. While chromatin remodeling is a highly conserved and well-characterized mechanism for the regulation of gene expression in Drosophila, the presence of cytosine methylation has been hotly debated for decades, where it has been reported to be absent, present at very low levels, or a rare event with unknown biological significance by different studies [58]. Given the uncertainty, the best approach to leverage Drosophila models for function exploration of tumor DNA methylation datasets would be to focus on genes whose expression levels are altered by DNA methylation in human tumors. Genetic modifier screens can be used to determine if altering the levels of Drosophila orthologs of these genes can modify tumor phenotypes.

## 6. Functional Exploration of Integrated Multi-Omics Analyses and Network-Based Models

Multi-omics integration studies and network-based models use sophisticated computational approaches to integrate data obtained by different high-throughput omics technologies [8,37,59]. These approaches explore interactions between different levels of alterations (DNA, RNA, protein, chromatin, and so on) to evaluate systems-level changes in tumor cells and identify integrated molecular signatures that correlate with different aspects of tumorigenesis, patient outcome, and drug response. These signatures are often complex and include genomic alterations or expression changes in multiple genes. Evaluating the functional relevance of these signatures in experimental tumor models may be challenging as they require simultaneous manipulation of a large number of genes and combining gain of function and loss of function approaches.

Another key advantage Drosophila brings as a functional validation platform for big cancer data is the ability to genetically manipulate multiple genes at once. Some of the tools developed for this purpose include the use of multigenic vectors, multiple short hairpin and guide RNA sequences, multicistronic expression systems that use self-cleaving peptides or Internal Ribosome Entry Sites (IRES), parallel genome editing, and multiplexable orthogonal and intersectional expression systems [16,17,60,61,62,63,64,65,66,67,68,69]. These tools can be used to build transgenic constructs that capture multiple gene expression changes observed in tumors, or those predicted as candidate biomarkers by computational approaches, and explore their roles in tumorigenesis. Such combinations can also be generated using standard genetic crosses and traditional meiotic recombination techniques; however, the number of genetic manipulations that can be captured using these methods will be more limited, particularly if the cancer model being used for validation studies is already genetically complex. Follow-up mechanistic studies can be used to explore emergent interactions between individual alterations within multigenic signatures to identify more specific hypotheses for mammalian validation studies.

## 7. Technical Limitations and Other Challenges

Challenges associated with functional exploration of big cancer data in experimental models can be broadly grouped in two categories: (1) achieving the genetic complexity required to capture tumor genomic landscapes in cancer models, and (2) being able to perform additional genetic manipulations needed for functional studies in these already complex genetic backgrounds. Tools and resources already available in Drosophila serve as an excellent starting point to address both of these challenges. For instance, our multigenic vector platform allows expression of two proteins and eight short-hairpins from a single multigenic construct [16]. This number can be increased to 4 proteins and 16 hairpins by bringing two multigenic constructs together in a single fly line by standard genetic crosses. If each hairpin is designed to target a different gene, up to 20 genes can be simultaneously manipulated using this approach.

It is also important to note that both the amount and the complexity of the big cancer data are growing at such a fast pace that increasing the number of genetic manipulations we can perform will not be sufficient to account for the genetic complexity of cancer. New methods that push the limits of Drosophila genetics even further in innovative ways will be necessary to increase our ability to simultaneously target different tissues or cell types in the same organism to build additional genetic complexity into our models.

In addition to technical challenges, it is also important to acknowledge that not all aspects of human tumors will be fully captured in Drosophila models. For instance, while most cancer relevant genes and signaling pathways are highly conserved in Drosophila, some genes altered in human tumors will not have Drosophila orthologs. For example, it may be challenging to model hormone-dependent tumors (breast, prostate) driven by alterations in nuclear hormone receptor activity in Drosophila. Nuclear hormone receptor function is broadly conserved in Drosophila, but there is significant divergence and specialization of functions and pathways [70]. Of note, there have been efforts to establish an estrogen-sensitive estrogen receptor (ER) expression system in Drosophila to study the interactions between ER and other signaling pathways [70,71,72]. It will be interesting to see if such an approach can be adopted to model hormone-dependent cancers of the breast and prostate.

Another limitation of Drosophila cancer models is the limited ability to capture and study tumor–stroma interactions. The tumor stroma is a complex environment that includes an extensive extracellular matrix, fibroblasts, immune cells, and vasculature, and impacts tumor progression and drug response [73]. For instance, even though the presence of an adaptive immune response with some parallels to the mammalian adaptive immunity has been reported in Drosophila [74], key immune cell types with known roles in tumor immunity are not present in flies. Of note, in recent years, there have been promising reports demonstrating that at least some aspects of the tumor microenvironment can be partially recapitulated in Drosophila cancer models [4,7], including some evidence of interactions between tumors and the tracheal system [75,76] and the presence of tumor associated blood cells (hemocytes) and cytokine signaling [7,77,78]. This is an important and expanding area in Drosophila cancer research; future studies will be crucial to explore what additional aspects of tumor-stromal interactions can be captured in Drosophila models.

As all model systems have their advantages and disadvantages, the most productive approach in any type of research is to play to the strengths of each experimental system. Therefore, it is important to recognize the limitations of Drosophila as a model system for cancer research and leverage its strengths to complement vertebrate models of cancer. We can achieve this by focusing our efforts on large-scale in vivo exploratory studies that embrace genetic complexity, an approach that may be particularly challenging in mammalian models and where Drosophila can make the biggest contribution to cancer research.

## 8. Concluding Remarks

Big cancer data are providing unprecedented access to global alterations in tumor cells at multiple levels and an impressive resolution. As our understanding of human tumor genomic and molecular landscapes becomes more sophisticated, there is an increasingly urgent need for genetically tractable cancer models to make sense of this large amount of data. With its sophisticated genetic tools and practical advantages, Drosophila offers a unique opportunity to address this important challenge in cancer research through genetic screens designed to functionally mine big cancer data and identify novel variants with cancer relevant roles. Big cancer data in return serve as a blueprint for building sophisticated and representative cancer models that are necessary for such genetics screens. Novel variants identified through such screens can then be incorporated into existing models to increase their genetic complexity in order to better capture the genomic landscape of cancer. Therefore, it is useful to think about the relationship between big cancer data and Drosophila cancer models as a dynamic and reciprocal one [79]; each novel variant identified and incorporated into a cancer model will alter its overall molecular landscape, bringing it one step closer to that of the original tumor. With each round of functional exploration and model building, more representative next-generation models that capture more of the genomic landscape of human tumors can be generated (Figure 1), increasing their utility and predictive power as cancer models and drug discovery platforms, providing opportunities to identify novel regulatory and druggable nodes that may be missed when simpler models are used.

As Drosophila researchers, we are a community of tool builders with a tradition of freely sharing reagents and resources. Large collections of publicly available transgenic and mutant lines that allow us to manipulate almost every gene in the Drosophila genome are a testament to this tradition. With our collective experience in technology development, our expertise in genetic modifier screens, and other complex in vivo genetic approaches that are routine in our laboratories, we are in a unique position to address this important challenge in cancer research brought upon by the postgenomic era.

## Figures and Tables

**Figure 1 ijms-21-03754-f001:**
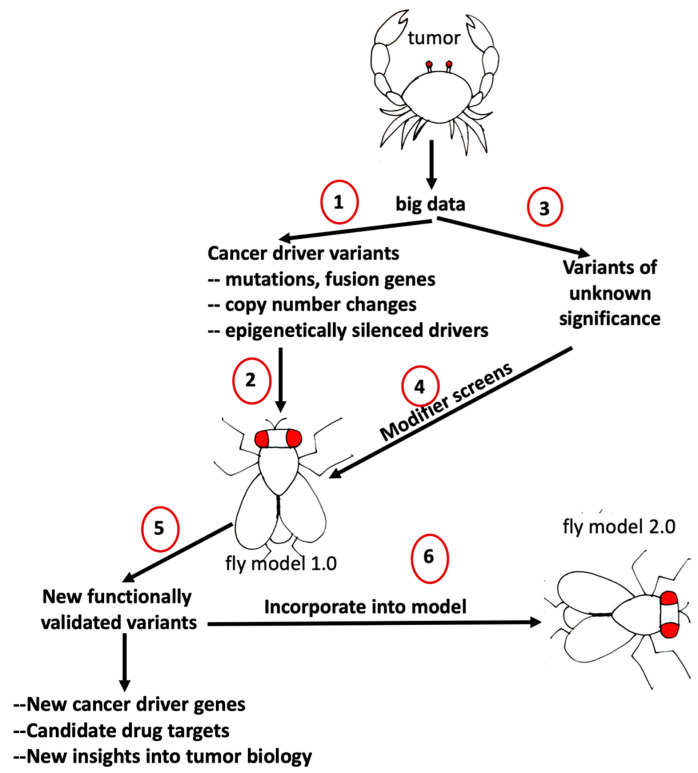
Functional exploration of tumor genome landscapes using tumor genome-based Drosophila cancer models. (1) A tumor genome dataset is mined to identify cancer driver mutations with previously established roles in tumorigenesis. (2) A Drosophila model that captures those alterations is established. (3,4) The Drosophila model used to screen variants with unknown significance to identify additional genes with potential roles in tumorigenesis. (5,6) Hits identified as modifiers of tumor phenotypes from the screen are incorporated into the Drosophila model to build a more complex and representative next generation model. Repeated rounds of model building and validation can be used to build increasingly complex models that better reflect the molecular landscape of sequenced tumors and find new variants.

**Figure 2 ijms-21-03754-f002:**
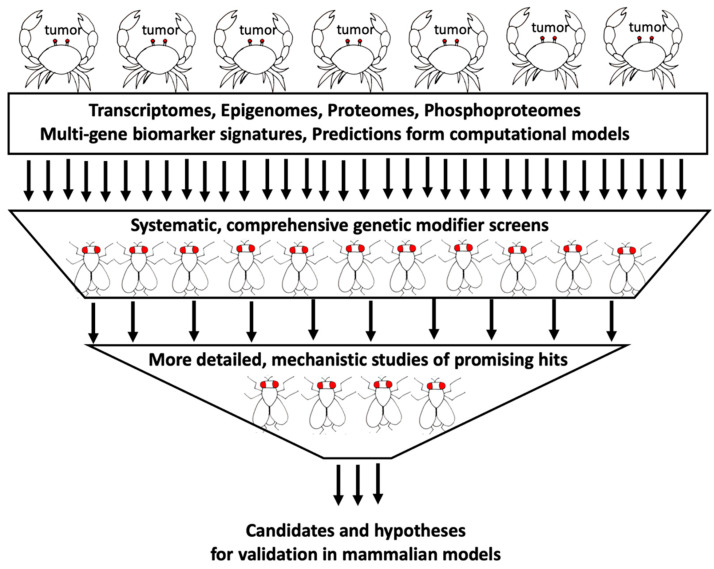
Strategy for functional exploration of different types of big cancer data. Comprehensive in vivo genetic modifier screens using tumor genome-based Drosophila models are performed to test functional relevance of gene expression changes, other alterations, and predictions from computational models. Validated hits are further explored in Drosophila to obtain insights into their mechanisms of action. This approach leverages genetic tools and practical advantages of flies in large-scale, exploratory studies to prioritize variants and establish more refined hypotheses to be tested in vertebrate models.

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
