# Peer review of "Strategies for Functional Interrogation of Big Cancer Data Using Drosophila Cancer Models"

_ijms, 2020, doi:10.3390/ijms21113754_

Round 1

Reviewer 1 Report

This is a very well written review that highlights the power of Drosophila for functional genetic tests of complex data from cancer genomics. Dr. Bangi is a good writer and it was overall a pleasure to read. I have a few suggestions that Dr. Bangi could consider.

A select use of more specific examples to support the different ideas would add substance to the narrative.

The main thesis of the review was testing hypotheses generated by cancer big data. The importance of past and present Drosophila developmental / cancer research for informing cancer mechanism was a bit undersold.  I suggest that direction should be stressed just a bit more for the naive reader, including the oncologist who doesn't know where the gene name "Notch" came from. 

Cite more recent  overview of TCGA , e.g. Ding et al. 2018 Cell 173:305.

Check reference list, some titles are missing words, e.g. #4 is missing "Drosophila"

Author Response

Comment: A select use of more specific examples to support the different ideas would add substance to the narrative.

Response: Since this is an opinion article discussing future strategies, it has been difficult to find specific examples from the literature to support each idea. I did find a few but I mostly added hypothetical examples illustrating how some of the strategies can be implemented (lines 91-95, 137-144, 265-274).

Comment: The main thesis of the review was testing hypotheses generated by cancer big data. The importance of past and present Drosophila developmental / cancer research for informing cancer mechanism was a bit undersold.  I suggest that direction should be stressed just a bit more for the naive reader, including the oncologist who doesn't know where the gene name "Notch" came from. 

Response: Two new paragraphs are added to the introduction to discuss the importance of Drosophila as a disease model in general and a cancer model specifically. I briefly summarized key milestones of Drosophila cancer research and referred the reader to some recent reviews that covered this topic more extensively.

Comment: Cite more recent  overview of TCGA , e.g. Ding et al. 2018 Cell 173:305.

Response: citation added. 

Comment: Check reference list, some titles are missing words, e.g. #4 is missing "Drosophila"

Response: Corrected and other references checked

P.S. Manuscript is also reorganized and Figure 1 changed to address comments from reviewer 2.

Reviewer 2 Report

This review article by Bangi explains interesting experimental strategies to use Drosophila for functional testing of “big data” related to cancer.  For the most part, it is well-written and represents a solid contribution to the literature; however, the descriptions and figures feel very vague, especially at the beginning, and so it would benefit overall from some re-organization and inclusion of more details.

The main suggestion I have is to bring up two points much earlier: 1) Discuss at the start the advantages of the Drosophila system (eg, what is covered in the paragraph starting at line 85).   Related to this, it needs to be clear that most human genes have a fly ortholog.  2) Since flies do not naturally get cancer, I think it is very important to explain explicitly what “cancer” looks like in Drosophila, how different cell types can be affected and give specific examples, and provide some genetic evidence as proof of principle that this is a valid model (eg, the tumor driver mutations in fly orthologs have similar cellular outcomes).  Some relevant characteristics come up at line 222. Explaining Drosophila as a model generally is necessary for understanding how big data candidate testing can added to the cancer model in a meaningful way.

The other issue that should be made clearer in the review is how the experiments will be assessed, and what the goals are. One of the subheadings is “From big cancer data to Drosophila models and back” but the going back part is not clear.  It seems at times like the author is saying the proposed strategies are advantageous just because they result in a more complex model – but more detail on what would happen next is needed.  He does suggest mammalian models could be developed, but couldn’t the fly models also be used for testing therapies? Or unbiased genetic screening? Similarly, how is it decided what is a “validated target” or “promising hit”? which genes would be selected to move forward with and why?

Minor points:

The figures could provide better representations of the underlying ideas.  Figure 1 is hard to follow in part because there are so many arrows. What happens with the “validated hits” are they just added to the next model? If so, what is the difference between arrow 5 and 6 (or 8 and 9)? What does “multiplexed orthogonal expression systems mean” in this figure? Maybe more definitions of the tools listed here would be helpful in general. It may help to add a figure with flow of the classification of subheadings (proteomics, epigenetics, transcriptomics) and how each could be carried forward methodologically. 

In sections 1-2, it wasn’t clear if the genetic changes discussed were in fly orthologs or human genes introduced into flies – this may be clearer with some changes in organization suggested above.

Cancer driver genes are defined on line 123 but should be defined when they are first mentioned.

Can the author provide some estimates for a limit on how many genetic changes can be included in one model/fly?

Related to the discussion of RNAs disrupted in cancer starting at line 227- these seem harder to model because I would expect that they may be less well conserved.  Do many/most human miRNAs have Drosophila homologs, and do they affect similar gene sets in both species?

The interaction between the genetic changes and the cell/tissue type comes up with respect to changes in chromatin (line 289), but wouldn’t this be relevant to many genetic changes?  If so maybe it can be discussed earlier in the review.

Author Response

Comment: The main suggestion I have is to bring up two points much earlier: 1) Discuss at the start the advantages of the Drosophila system (eg, what is covered in the paragraph starting at line 85).   Related to this, it needs to be clear that most human genes have a fly ortholog.  2) Since flies do not naturally get cancer, I think it is very important to explain explicitly what “cancer” looks like in Drosophila, how different cell types can be affected and give specific examples, and provide some genetic evidence as proof of principle that this is a valid model (eg, the tumor driver mutations in fly orthologs have similar cellular outcomes).  Some relevant characteristics come up at line 222. Explaining Drosophila as a model generally is necessary for understanding how big data candidate testing can added to the cancer model in a meaningful way.

Response: Two new paragraphs are added to the introduction to discuss the importance of Drosophila as a disease model in general and a cancer model specifically. I briefly summarized key milestones of Drosophila cancer research and referred the reader to some recent reviews that covered this topic more extensively.

Comment: The other issue that should be made clearer in the review is how the experiments will be assessed, and what the goals are. One of the subheadings is “From big cancer data to Drosophila models and back” but the going back part is not clear.  It seems at times like the author is saying the proposed strategies are advantageous just because they result in a more complex model – but more detail on what would happen next is needed.  He does suggest mammalian models could be developed, but couldn’t the fly models also be used for testing therapies? Or unbiased genetic screening? Similarly, how is it decided what is a “validated target” or “promising hit”? which genes would be selected to move forward with and why?

Response: The manuscript is reorganized to address the reviewer's overall concern, clarify the goals and the approach. New paragraphs discussing what type of assays can be used in such screens and follow up studies are added (lines 129-146). The section titled “From big cancer data to Drosophila models and back” is deleted and the advantages of of complexity are more clearly explained (see concluding remarks).Terms like "validated target" and "promising hit" are removed for clarity.

Flies can certainly be used for  drug screening or unbiased genetic screens but such uses are beyond the scope of this article. The focus  here is exploring the functional relevance of genomic changes observed in sequenced human tumors. So the strategies discussed here focus on targeted screens guided by genomic data. 

Comment:The figures could provide better representations of the underlying ideas.  Figure 1 is hard to follow in part because there are so many arrows. What happens with the “validated hits” are they just added to the next model? If so, what is the difference between arrow 5 and 6 (or 8 and 9)? What does “multiplexed orthogonal expression systems mean” in this figure? Maybe more definitions of the tools listed here would be helpful in general. It may help to add a figure with flow of the classification of subheadings (proteomics, epigenetics, transcriptomics) and how each could be carried forward methodologically. 

Response: Figure 1 was originally intended to demonstrate how fly models can be updated by incorporating novel variants shown to be have important roles in tumorigenesis in prior screens to bring them closer to the molecular complexity of the human tumors that they model. Such models would better reflect the molecular landscapes of human tumors. Findings from these models would be more likely to be relevant for  human cancer. This figure is now simplified by removing additional arrows and multiple rounds of screens and model building. The term “multiplexed orthogonal expression systems” is removed from the figure and more simply defined in the main text as methods that allow simultaneous targeting of multiple tissues or cell types in the same organism (line 394)

Comment: In sections 1-2, it wasn’t clear if the genetic changes discussed were in fly orthologs or human genes introduced into flies – this may be clearer with some changes in organization suggested above.

Response: In the introduction, it is now stated that fly models have been successfully generated both ways. Strategies discussed for genetic screens include both manipulating Drosophila orthologs and introducing human versions. These are described in detail in section 2: "Functional validation of genomics datasets"

Comment: Cancer driver genes are defined on line 123 but should be defined when they are first mentioned.

Response: This is now addressed by the reorganization

Comment: Can the author provide some estimates for a limit on how many genetic changes can be included in one model/fly?

Response: Discussed in lines 385-389

Comment: Related to the discussion of RNAs disrupted in cancer starting at line 227- these seem harder to model because I would expect that they may be less well conserved.  Do many/most human miRNAs have Drosophila homologs, and do they affect similar gene sets in both species?

Response: It is clarified that for genomic alterations with broad, pleiotropic effects like RNA's, the focus would be on exploring the roles of the genes targeted by them using Drosophila models (line 257-274). The reviewer is correct that even if the RNAs are conserved, their target profiles are unlikely to be identical in flies. By focusing on exploring the roles for their targets, we can still use Drosophila in a productive way to investigate the mechanisms of action for these RNA's

Comment: The interaction between the genetic changes and the cell/tissue type comes up with respect to changes in chromatin (line 289), but wouldn’t this be relevant to many genetic changes?  If so maybe it can be discussed earlier in the review.

Response: It is now explained early on in the introduction that Drosophila cancer models have been successfully established by both targeting corresponding fly tissues and generic epithelia where all cancer relevant signaling pathways are conserved. While it may be better to target the corresponding cell/tissue type, more generic Drosophila cancer models have also been extremely useful.